# Bringing GPCR Structural Biology to Medical Applications: Insights from Both V2 Vasopressin and Mu-Opioid Receptors

**DOI:** 10.3390/membranes13060606

**Published:** 2023-06-16

**Authors:** Aurélien Fouillen, Julien Bous, Sébastien Granier, Bernard Mouillac, Remy Sounier

**Affiliations:** 1Institut de Génomique Fonctionnelle (IGF), Université de Montpellier, CNRS, INSERM, 34000 Montpellier, France; aurelien.fouillen@igf.cnrs.fr (A.F.); sebastien.granier@igf.cnrs.fr (S.G.); bernard.mouillac@igf.cnrs.fr (B.M.); 2Centre de Biochimie Structurale (CBS), Université de Montpellier, CNRS, INSERM, 34090 Montpellier, France; 3Section of Receptor Biology & Signaling, Department of Physiology & Pharmacology, Karolinska Institutet, 17165 Stockholm, Sweden; julien.bous@ki.se

**Keywords:** G protein-coupled receptor, μ-opioid receptor, arginine-vasopressin receptor, Nuclear Magnetic Resonance, cryo-electron microscopy, X-ray crystallography, molecular dynamics

## Abstract

G-protein coupled receptors (GPCRs) are versatile signaling proteins that regulate key physiological processes in response to a wide variety of extracellular stimuli. The last decade has seen a revolution in the structural biology of clinically important GPCRs. Indeed, the improvement in molecular and biochemical methods to study GPCRs and their transducer complexes, together with advances in cryo-electron microscopy, NMR development, and progress in molecular dynamic simulations, have led to a better understanding of their regulation by ligands of different efficacy and bias. This has also renewed a great interest in GPCR drug discovery, such as finding biased ligands that can either promote or not promote specific regulations. In this review, we focus on two therapeutically relevant GPCR targets, the V2 vasopressin receptor (V2R) and the mu-opioid receptor (µOR), to shed light on the recent structural biology studies and show the impact of this integrative approach on the determination of new potential clinical effective compounds.

## 1. G Protein-Coupled Receptor (GPCR) Drug Targets

G protein-coupled receptors (GPCRs) are the largest superfamily of cell surface signaling membrane proteins that comprise up to 826 molecules in humans [1]. This superfamily can be divided into six main classes—A for Rhodopsin; B with B1 for Secretin and B2 for adhesion; C for Glutamate; D1 for a Ste2-like fungal pheromone; F for Frizzled; T for Taste 2—based on the phylogenetic information or their amino acid sequences [2]. GPCRs are complex allosteric machines with high flexibility and comprise a bundle of seven transmembrane domains (TM) [1]. Their ability to adopt different conformations allows GPCRs to respond to various extracellular stimuli (light, ions, forces, peptides, lipids, and proteins) by transmitting their signal across the membrane to activate diverse intracellular signaling pathways [3,4]. The high conformational flexibility of GPCRs is thus a prerequisite for sensing a variety of stimuli and links them to different signaling partners such as G proteins and arrestins [5] (Figure 1). However, this high plasticity represents a challenge for structure determination. Once activated by external factors, GPCRs elicit diverse second messenger-mediated pathways through their coupling to their signaling partners, such as calcium mobilization, the phosphorylation process (i.e., of extracellular regulated protein kinase 1/2 (pERK1/2)), or cyclic adenosines-3,5-monophosphate (cAMP) [6]. These diversified downstream signaling pathways triggered by GPCRs make them attractive for drug development for a wide range of therapeutic areas, including inflammation and diseases of the central nervous system as well as the cardiovascular, respiratory, and gastrointestinal systems [7]. Indeed, GPCRs are among the most important drug targets for many human diseases, with 527 FDA-approved drugs and 60 drug candidates targeting them [7,8].

Most drugs have been discovered using traditional empirical methods, such as high-throughput screening (HTS) [9,10,11,12,13]. These methods require the screening of thousands of compounds based on phenomenological observations in recombinant or endogenous systems. The obtained hits are then optimized by structural activity assays. Unfortunately, many of the available compounds can display important side effects in addition to poor selectivity. It is, therefore, necessary to find more rational ways to design selective drugs with few or no side effects. Determining the three-dimensional (3D) structures of biological molecules is critical to a better understanding of proteins’ complexity and their biochemical features. It links them to their biological roles and ultimately allows structure-based drug design (SBDD). The increasing number of GPCR structures in complex with different ligands and/or signaling partners available in the Protein Data Bank (PDB) has provided an unprecedented opportunity for computational drug discovery [14,15]. SBDD allows the design and optimization of rational drugs based on millions of molecules that are docked to the structure of their biological target. Molecules derived from SBDD provide an easier path toward the optimization of affinity and receptor selectivity. Indeed, the SBDD approach has recently demonstrated successful results in the discovery of new molecules after docking over millions of molecules to the mu-opioid receptor (µOR) binding pocket [16], as well as to allosteric sites of the M2 muscarinic receptor (M2R) [17].

This review aims to present an updated knowledge of the structural tools that have enabled the structural and dynamic characterization of GPCRs based on the case of two widely drug-targeted class A GPCR receptors: the µOR and the arginine-vasopressin (AVP) V2 receptor (V2R).

## 2. Structural Development Methods to Characterize GPCRs

The determination of the three-dimensional (3D) structures of biological molecules is critical to better understanding the complexity of proteins and their biochemical features. It links them to their biological roles and ultimately allows drug design. Currently, three main methods—X-ray crystallography, Nuclear Magnetic Resonance (NMR) spectroscopy, and Cryo-Electron Microscopy (cryo-EM)—are used to structurally characterize molecules at a high resolution (Figure 2). First, X-ray crystallography has been extensively used to identify the 3D structures of both small molecules and large protein complexes. However, several macromolecules or complexes cannot crystallize due to multiple limiting factors, such as the presence of flexible regions, their solubility, pH sensitivity, or temperature stability. Secondly, NMR can provide high-resolution structural information, especially on the disordered regions of a given molecule. However, NMR is typically used for the structural determination of small proteins and becomes challenging for proteins larger than 50 kDa. Nevertheless, NMR provides crucial information about dynamics and binding properties. Finally, cryo-EM applies to proteins with sizes larger than approximately 60 kDa and allows for the determination of proteins in vitrified states. For many years, X-ray crystallography was the most used structural method to reach an atomic resolution. It is a powerful technique that can solve the structure of soluble proteins, but it remains challenging for membrane proteins despite recent advances. Over the last decade, single-particle cryo-EM has experienced an important revolution with the development of direct electron (DE) detectors and image-processing software. These technological developments have redefined the size limit of the biological samples studied. Indeed, before the multiple developments of EM detectors, microscopes, and processing software, it was quite difficult to observe samples under 200 kDa. Currently, it is possible to structurally characterize smaller specimens, such as the small hemoglobin protein (64 kDa) [18] and the streptavidin (52 kDa) [19] at a resolution of 3.2 Å. Cryo-EM is thus becoming a method of choice for the structural determination of small complexes and, consequently, has become an alternative for the structural characterization of membrane proteins [20].

Remarkable progress in the structural characterization of GPCRs has led to a better understanding of these signaling key players (Figure 3). Indeed, since the first structure of rhodopsin in 2000 [21] and of the β2-adrenergic receptor (β_2_AR) in 2007 [22], the number of GPCR structures has increased almost exponentially, ushering a GPCR structural revolution that has brought new insights into GPCR biology and renewed interest to GPCR drug discovery. Such a structural characterization has been due to the improvement of (a) GPCR expression and purification (which have been engineered to increase their stability and production level, and the development of the lipid cubic phase) and (b) advances in technologies, such as for X-ray crystallography (micro-focus X-ray sources at synchrotrons and X-ray free electron lasers) and (c) cryo-EM (DE detectors, better microscopes, faster data collection and image processing). These improvements have led to the current determination of 766 class A GPCRS structures from 120 different receptors, allowing the determination of diverse states from multiple inactive states, active-intermediate states, and active states in complex with either G proteins or arrestins (for the current list of resolved GPCR structures, please see the GPCRdb [23]). One of the most groundbreaking discoveries in our understanding of GPCR biology was the first high-resolution view into a GPCR that was bound to its signaling partner obtained by X-ray crystallography [24], namely the agonist-bound β_2_AR-Gs complex structure (Figure 3).

While this achievement has been performed by X-ray crystallography, cryo-EM then led to a surge of structures of GPCR coupled to signaling molecules. Most of those were structures of GPCRs linked to G proteins, but recently, a handful of GPCR–arrestin complexes have also been described [25,26,27,28]. Indeed, GPCRs are generally subjected to conformational rearrangements at the intracellular face of the TM bundle [29]. Both arrestins and G proteins bind with a high degree of shape that is complementarity to a cavity in the cytoplasmic surface of the receptor and created by the outward movement of TM6. This cavity accommodates the α5-helix of the G_α_ subunit or the finger loop of arrestins [30]. Although arrestins bind in the same pocket as G proteins, they form a larger interface with these receptors, which involves contacts with intracellular loop1 (ICL1) that are not observed in G protein complexes. Even though arrestins and G proteins interact with common regions of GPCRs (intracellular ends of TM3, TM5 and TM6, and intracellular loop2 (ICL2)), the organization of these interfaces differs significantly (Figure 1). In addition, the C terminal portion of GPCRs interacts tightly with the N-lobe of arrestins: a feature that is not observed with G proteins. Despite similarities between the active-state structures of rhodopsin, β_2_AR, M2R, μOR, adenosine A_2_AR, calcitonin receptor (CTR), and glucagon-like peptide 1 receptor (GLP1-R), which have been solved by X-ray crystallography or by cryo-EM, there have also structural differences [29]. To fully understand the molecular mechanisms of these complex types of machinery, it has been necessary to obtain not only atomic resolution models but also information about their dynamics. For a given receptor, it is, therefore, important to solve at least several states: such as the atomic structure of (a) the active agonist-receptor-G protein complex, (b) the active agonist–receptor-arrestin complex, (c) an inactive state (bound to an inverse agonist or an antagonist), and (d) an apo-state.

Even with these milestone studies, defining the active states of a given unresolved GPCR complex is still a difficult challenge due to difficulties in the biochemistry of GPCRs, the dynamics and the size of these systems, and the highly sophisticated mechanism of GPCR activation. To obtain a complete drawing of the activation of a given receptor, it is necessary to either (a) capture multiple snapshots of a given state or (b) use complementary methods such as NMR or molecular dynamics (MD) to understand the whole mechanism that implies the flexibility and adaptability of the GPCR. NMR spectroscopy in solution has provided fundamental insights into GPCR’s dynamic, allostery, and conformational rearrangements [31]. While allowing a deep investigation into GPCRs, the characterization of large membrane proteins remains challenging by NMR. However, NMR studies on limited receptor regions such as the intracellular loop 3 (ICL3) [32] and also the C-terminal domain [33] have been instrumental to our understanding of partner interactions. Furthermore, numerous studies have successfully characterized GPCR by using site-specific isotopic labeling after the expression by either introducing a ^13^C-methyl group on exposed Lysine [34] or by incorporating ^19^Fluorine to label Cysteine [35]. While these methods are not trivial and require the extensive production of GPCRs, they have been used to characterize several GPCRs. However, on large proteins such as GPCRs, the abundance of ^1^H atoms severely affect the signal-to-noise ratio. The use of deuteration can minimize the abundance of hydrogen atoms despite its expensive cost [36]. Additionally, to choose the right labeling strategy, it is important to determine the number and distribution of either cysteine, lysine, or methionine residues. Indeed, to gain more structural information, it is required that as many probes are labeled as possible. On average, 10 to 15 residues are available for potential labeling on GPCRs, which is limiting for data analysis. Despite these limitations, solution NMR has been prevalent in revealing novel insights into the dynamics and conformations of receptors. This experimental approach has been recently associated with molecular dynamic (MD) simulations to converge toward more accurate models. MD is a computer simulation strategy that can analyze the physical movements of atoms and, thus, ultimately of molecules. Due to an incredible improvement in computing performances and MD methods, simulations now generate more and more realistic dynamic ensembles [37]. Those simulations can predict, among others, conformational changes, protein folding, and ligand binding. This dynamic information is particularly interesting when coupled to experimental data such as (a) NMR, which validates the binding poses of a ligand or the conformation of a protein, and (b) X-ray crystallography and cryo-EM which bring complementary data on the subtle movements that occur at the protein level of the observed frozen-states.

While our introduction focuses on cryo-EM, X-ray crystallography, NMR, and MD to deeply characterize a given protein, several approaches can also be considered to add crucial information that directly observes the conformational ensemble of GPCRs at all protein levels (i.e., transition kinetics of different conformations, interaction); this has been shown for Double electron-electron resonance (DEER) [38], Fluorescence or Bioluminescence resonance energy transfer (BRET/FRET) [39,40], electron paramagnetic resonance (EPR) [41,42], Hydrogen/Deuterium exchange mass-spectrometry (HDX-MS) [43] and also supported by molecular dynamics (MD) simulations [37,44]. However, these complementary approaches are not discussed further in this review.

## 3. µOR and V2R: Two Representative Peptide Class A GPCRs

The µORs are integral membrane proteins that are widely distributed throughout the central nervous system and the periphery; they are involved in many brain and behavioral functions, including pain, analgesia, motivation, reward, and addictions [45]. µOR is the main pharmacological target for the management of moderate to severe pain. µOR are activated by a structurally diverse spectrum of natural and synthetic opioids, including endogenous endorphin peptides, morphine, and fentanyl. The signaling of µOR is primarily transduced through heterotrimeric G proteins, particularly pertussis toxin-sensitive Gi/o proteins. The release of the G_α_ subunit inhibits adenyl cyclase (AC), and the release of G_βγ_ subunits activates K^+^ channels and inhibits voltage-gated Ca^2+^ channels, with AC-dependent decreases in cAMP levels being the most direct and immediate cellular event [46]. Opioid receptors, similar to most GPCRs, are desensitized by the action of (a) GPCR-regulated kinases (GRKs) that phosphorylate receptors and (b) β-arrestins, which internalize phosphorylated receptors [45]. The dynamic and localization aspects of opioid receptor interactions with these proteins, and numerous other intracellular partners, gives rise to varied signaling outcomes. It has been proposed that opioid-induced analgesia originates from Gi signaling, whereas unwanted side effects such as respiratory suppression and constipation might be caused by β-arrestin signaling, as reviewed in [47]. However, recent findings argue that G protein selectivity could improve analgesia and tolerance but not necessarily the side effects [48]. Opioid-induced respiratory depression and constipation could be independent of β-arrestin signaling [49]. Despite G protein-selective µOR agonist hypothesis being increasingly controversial, their biological complexity continues to provide hope that an ideal opioid analgesic, one devoid of the lethal and addictive properties of most opioids, may exist. To achieve this, a better understanding of the receptor’s structural pharmacology is needed as this could allow for the rational design of opioids with the desired specificity and efficacy.

The arginine-vasopressin (AVP) is a nine-amino acid peptide hormone, which mainly acts in the mammalian kidneys by regulating the body’s water balance and solute transport by interacting with the vasopressin V2 receptor (V2R) [50], but also possessing important physiological effects in the whole body, such as a vasodilatory role in the endothelium [51] or the hydraulic pressure control of the endolymphatic system in the inner ear [52]. The V2R preferentially couples to the Gs protein resulting in the activation of adenylyl cyclase [50]. It is also the most used model in the field to study the molecular pharmacology of arrestins’ recruitment to GPCRs. The dysfunctions of this GPCR result in clinical disorders from the dysregulation of water balance resulting in the syndrome of inappropriate antidiuretic hormone secretion (SIADH, as a consequence of many forms of cancer [53]), congestive heart failure, hepatic cirrhosis, and urine disorders (incontinence, nocturia) [54]. V2R is also a target for treating autosomal dominant polycystic kidney disease (PKD): the most frequent Mendelian inherited disorder affecting millions of people worldwide [55]. Half of PKD patients over sixty require dialysis or kidney transplantation. Unfortunately, as of today, there is no specific drug that reduces side effects on the market to treat disorders linked to V2R. In this context, deciphering the atomic structures of the different states of V2R when bound to molecules is suggested to offer potential clinical benefits [56] and represent a great opportunity for the development of therapeutic molecules.

## 4. Solving High-Resolution GPCRs Structures by X-ray Crystallography and Cryo-Electron Microscopy

### 4.1. X-ray Crystallography to Investigate µOR

As mentioned before, X-ray crystallography determined the first-ever GPCR structure at a high-resolution [21]. Initially, receptors were usually fused to protein modules (i.e., T4 Lysozyme (T4L), cytochrome b562 RIL (BRIL)) to improve their chance of contact formation and crystal generation. At this point, the system with the highest likelihood of success was a GPCR with a high-affinity antagonist. Indeed, the potential to maintain GPCR that was inactive and thus in a rigid conformation was suitable for crystallogenesis and high-resolution diffraction. For a more extensive view of technological advances in the determinants used for GPCR crystallography, please refer to [57]. Based on these advances, the first structure of the µOR was obtained by inserting a T4L fusion protein module into the ICL3 [58]. The receptor was stabilized in the inactive conformation by the addition of an irreversible morphinan agonist β-funaltrexamine (β-FNA) with the potential to be covalently bound with the K^5.39^ (Ballesteros and Weinstein’s nomenclature in superscript, consisting of two numbers where the first denotes the helix ((one to seven)), and the second denotes the residue position considering the reference to the highly conserved amino acid of the considered helix [59]). Interestingly, compared to the β_2_AR bound antagonist structure [22], β-FNA was more exposed to the extracellular solvent, in addition to a deep insertion in the receptor. The authors hypothesized that this peculiar feature might explain the dynamic binding of µOR antagonists, which feature a fast dissociation rate despite their high affinity. Indeed, it appears that the ligand interaction is mainly driven by hydrophobic contacts: one covalent bond, and three polar contacts, involving D^3.32^ with a β-FNA tertiary amine (Figure 4a), and both H^6.52^ and Y^3.33^ with aromatic rings of the morphinan group. Since dimerization has been proposed to play a role in receptor trafficking and morphine tolerance [60], it is interesting that the crystal oligomeric arrangement shows a large oligomeric interface involving TM5 and TM6 with a surface area of 1492 Å^2^ for each receptor. Furthermore, based on cellular data that demonstrate the presence of homo- and heterodimers in living cells by quantitative-BRET [61,62], this unexpected interaction could be physiological.

Following the inactive structure of µOR when bound to β-FNA and obtained by X-ray crystallography, an active conformation was investigated using a high-affinity agonist. The morphinan agonist BU72 [63] and a G-protein mimetic nanobody (Nb39) were used to stabilize µOR in its active conformation [64]. Nb39 was generated by immunizing the llama with purified µOR in complex with an agonist. These generated nanobodies were tested by competition assays with BU72. Crystals of BU72-µOR-Nb39 obtained by LCP were measured and provided a dataset with a resolution of 2.1 Å (Figure 5a). Firstly, when compared to its inactive counterpart, the structure displayed common GPCR activation features, such as a large outward motion of the TM6 and the inward motion of the TM5 and TM7 to a lesser extent. Interestingly, on the extracellular side of the receptor, the authors point out that there were few structural differences between the active and inactive structures. The amino-terminal residue H54^N-ter^ appeared to interact with BU72, acting similar to a cap, but this interaction was not critical for the agonist’s affinity and was not observed in the following cryo-EM structure [65], but was by NMR with different agonists [66]. The BU72 interacts in a similar position and orientation as the antagonist β-FNA. This interaction was driven by hydrophobic interactions as well as by two polar contacts. These included a water network interaction involving the phenolic hydroxyl of BU72 and H^6.52^ and an ionic interaction involving the tertiary amine of BU72 and D^3.32^ (Figure 4e). These two ionic interactions were also present in the antagonist’s structures. Interestingly, a substitution of the BU72 tertiary amine methyl with a cyclopropylmethyl triggered an antagonistic effect [67]. While the ligands shared the same scaffold, subtle differences in their position induced small rearrangements in the TM3 and TM6: a rotameric change in the sodium coordinating residue N^3.35^, as well as the reorganization of the water-mediated polar network. This subsequently triggered the propagation of conformational rearrangements of the conserved motifs in class A GPCRs, such as the W^6.48^ toggle switch and P^5.50^I^3.40^F^6.44^, N^7.49^P^7.50^xxY^7.53^, and D^3.49^R^3.50^Y^3.51^ motifs. This allosteric process induced cytoplasmic rearrangements, such as the large outward movement of TM6, which was necessary to accommodate G protein binding. Interestingly, in these two publications, X-ray crystallography was instrumental in unraveling the details of µOR’s function. Indeed, the data provided valuable information on both the active and inactive states, allowing not only the binding of the respective ligands to the µOR to be characterized alongside its architecture but also to partially delineate µOR activation and understand how agonists and antagonists fulfill their respective roles by interacting with different sites and inducing different conformational modifications.

### 4.2. Cryo-EM on µOR G-Protein Activated Structures

Despite the X-ray structures discussed above and their great importance for our understanding of the overall structural changes during activation, it was necessary to wait for the first cryo-EM structure to be able to observe the µOR in its active state and coupled with its main signaling partner [65]. Indeed, the first cryo-EM structure of the µOR was bound to both the peptide agonist DAMGO [68] and a nucleotide-free heterotrimeric Gi protein (Gi). The complex was stabilized by the single-chain variable fragment 16 (scFv16) (Figure 5b). Interestingly, despite their different chemical nature and flexibility, DAMGO and BU72 were superimposed in the µOR’s orthosteric pocket (Figure 4b,e). While DAMGO extended further toward the extracellular side, both agonists interacted with the same polar contacts. The µOR D^3.32^ interacted with the N-terminal extremity of DAMGO, and an MD simulation revealed the same water-mediated interaction with H^6.52^. This MD simulation also indicated a potential transient contact between DAMGO and the µOR Y^7.43^. Furthermore, this structure highlighted the role of extracellular loops (ECLs) in an opioid receptor subtype specificity, which was consistent with previously published results using δ/µOR and κ/µOR chimera to investigate DAMGO selectivity toward µOR [69]. Gi protein and Nb39 stabilized structures exhibited the same activation features discussed above, including the reorganization of D^3.49^R^3.50^Y^3.51^, N^7.49^P^7.50^xxY^7.53^, and P^5.50^I^3.40^F^6.44^ motifs. Furthermore, both Gi and Nb39 induced a similar increase in their agonist affinity [64], which is consistent with the concept of a ternary complex with bidirectional allosteric cooperativity, and is accepted in the field of GPCRs. The only difference was a slightly larger outward motion of the TM6 and a different conformation of ICL3. The µOR–Gi interface was consistent with the general G proteins–GPCR interfaces, which have a conserved overall architecture [70]. The C-terminal Gαi1 α5 helix made most of the contact due to its insertion into a pocket from the transmembrane core. Additional contacts to stabilize the complex were between the ICL2, ICL3, TM3, TM5, and TM6 with the RAS-like domain of Gαi1 [65].

µOR is the main target for opioid analgesics, and current drugs exhibit detrimental side effects such as addiction, tolerance, and respiratory depression [71], making it a prime target. The delineation of biased agonisms, the understanding of distinct signaling profiles, and the application of a structural-based drug design (SBDD) to find better drugs with fewer side effects have been long-standing goals in the field of GPCR structural biology with limited success. Recently, four publications have explored these more subtle structural concepts for the µOR [72,73,74,75].

Indeed, to better understand the relationship between the ligand activation mechanism and physiological response, a study [72] investigated the structures of the G protein-biased partial agonist, PZM21 [16], which was discovered by SBDD based on the inactive µOR structure, and was shown to exhibit a reduction in side effects compared to conventional agonists such as morphine. The authors used this finding to design a new and more biased agonist: FH210. They successfully solved the structures of µOR-miniGi when bound to either PZM21 or FH210 to validate their binding pose (Figure 4d). Interestingly, the µOR-miniGi stabilized cryo-EM structures were very similar to the DAMGO-µOR-Gi structure [65]. The three ligands overlapped in the catalytic pocket with the same orientation and made the same sets of interactions with D^3.32^ and H^6.52^. Since DAMGO ascended further toward the N-terminal side, PZM21 and FH210 both made contact in different ways, with FH210 displaying a better complementarity with the receptor. They nonetheless differed in their interaction with the ends of TM2 and TM3 and with ECL1 (proposed to be involved in subfamily specificity). This subtle difference was hypothesized to be responsible for the biased agonism toward Gi. Furthermore, there was no difference in the biased and unbiased agonist structures in terms of activation and the µOR-G–protein interface [72].

A second study investigated the mechanistic basis of action for lofentanil (LFT) and mitragynine pseudoindoxyl (MP): two agonists with different profiles [74]. LFT is associated with a higher risk of respiratory dependence and overdose, whereas MP is a µOR G-protein biased agonist with reduced side effects compared to other opioids. Cryo-EM structures of the µOR–Gi complex with MP and LFT revealed that both ligands engaged in different subpockets (Figure 4f,g). They partially shared interactions with D^3.32^ but not with H^6.52^, as previously described for DAMGO, BU72, PZM21, and FH210. Instead, LFT and MP dived deeper into the pocket, making a series of hydrophobic contacts. However, these two molecules differed due to their interactions with the top TM1, TM2, TM3, and TM7 and accommodated two different subpockets, which were separated by Q^2.60^ and featured different µOR side chain orientations, with LFT and MP contacting TM2, TM3, ECL2, TM2, TM3 and TM7, respectively. Complementary MD simulations displayed a potential interaction between Q^2.60^ and Y^7.43^, which is frequently found in LFT-bound simulations, occasionally found in DAMGO-bound simulations, and not or rarely found in MP-bound simulations. MD simulations allowed the extrapolation of the effect of these three ligands with distinct pharmacological profiles on µOR cytoplasmic side conformation. Interestingly, MP and LFT seemed to favor two distinct conformations, with MP inducing a canonical state similar to the one induced by G protein binding, and LFT led to an alternative state with a rotation in the TM7 and a relaxed N^7.49^P^7.50^xxY^7.53^ motif compared to the canonical one. DAMGO promoted an equilibrium between these two conformations. These conformational differences could explain the difference in recruitment profiles and, subsequently, the agonist-dependent physiological profiles.

To facilitate the rational design of next-generation analgesics, the structures of the human µOR–G protein complexes being bound to morphine, fentanyl, oliceridine, PZM21 and SR17018 have been recently reported [73]. Interestingly, all three biased G-protein ligands, SR17018, oliceridine and PZM21, were found to bind to the orthosteric binding pocket just above W^6.48^ (Figure 4c,d,g). Fentanyl adopted a Y-shaped conformation in the orthosteric binding pocket, similar to LFT [74]. The phenylethyl moiety was faced toward the cleft of TM2 and TM3, the propionyl group faced TM6, and the n-aniline ring faced TM7, where they all formed hydrophobic interactions. Oliceridine adopted a similar binding pose to fentanyl, with the pyridine ring of olicerinide forming hydrophobic interactions with TM6/7. Both SR17018 and PZM21 adopted conformations away from TM7, and only weak hydrophobic interactions took place with TM6 residues. Compared to other ligands, morphine interacted with hydrophobic residues of TM3, TM6, and TM7. The morphinan group overlapped with the tyrosine of DAMGO, with the hydroxyl of the phenol moiety of morphinan pointing toward TM5. The amine groups in the tyrosine of DAMGO and in the morphinan group of morphine formed salt bridges with the carboxylate group of D^3.32^, which is a universal interaction of ligands with opioid receptors, as seen with the piperidine ring of fentanyl.

In the same idea, another study used SBDD to develop bitopic ligands targeting both the orthosteric binding site with a fentanyl scaffold and the class A conserved sodium (Na^+^) binding site to Gi biased agonists [75] with a positively charged guanidino group (Figure 4h). Indeed, in a physiological concentration, Na^+^ was demonstrated to reduce µOR activation [76,77,78] and could bias GPCR profiles toward either G-protein or arrestin pathways. The lead bitopic fentanyl derivatives, C5 and C6 guano, maintained high potency and efficacy at Gi subtypes and showed reduced arrestin recruitment compared to fentanyl while displaying an efficacy equivalent to morphine. Interestingly, different ligands feature variable efficacy and potency, values, and selectivity, between Gi/o/z subtypes and arrestins. Since G proteins activate specific pathways with diverse associated cellular responses, including unwanted effects, modulating and understanding these bitopic ligand-dependent G protein subtype’s selectivity might be key to developing new kinds of analgesics with fewer side effects [75].

However, since all these structures are G protein stabilized complexes, the G protein certainly stabilized the µOR intracellular side in a specific conformation independent of the variable conformational changes induced by different agonists on an unbound and less constrained receptor and in a more physiological context. Therefore, the signal transduction process leading to β-arrestin recruitment remains to be elucidated. Indeed, to elucidate how ligand binding leads to different intracellular conformations, resulting in different coupling effects of µOR to the G protein compared to β-arrestin, other techniques, such as NMR presented below, are required.

### 4.3. Cryo-EM, the Case of V2R

Similar to many other GPCRs, V2R has resisted X-ray crystallography despite extensive research worldwide. This is probably due to its strong dynamics, which have so far prevented the obtention of well-ordered crystals that are necessary for X-ray crystallography.

It was only with the advent of cryo-EM as a reference method for the high-resolution characterization of GPCRs that the molecular details of the active structure of V2R bound to its natural hormone, the arginine vasopressin (AVP), in complex with the Gs protein was first experimentally observed (Figure 6a) [79,80,81]. Interestingly, the different strategies used by these three groups have allowed them to extract different and complementary information. Indeed, while in one case [79], the system consisted of a wild-type Gαs protein and few modifications to keep the protein complex as physiological as possible, the other two publications [80,81] used engineered systems (miniGsGi chimera and/or nanobit tethering strategy [82]) to reduce these dynamics and thus favored a stable complex. These structures were shown to share a common architecture in which the AVP adopts a central pose in the TMs bundle and interacts with all TMs and ECL1 and ECL2. The cyclic component of the nonapeptide (Cys1 to Cys6) goes deeper into the orthosteric site and makes contact with the core of the pocket, while the linear part (Pro7 to Gly9-NH2) adopts a shallow position with more potential flexibility and interactions with ECL1 and ECL2. In all cases, V2R was shown to be an active conformation featuring a large outward motion of TM6 and minor changes in TM5 and TM7 compared to the inactive oxytocin receptor structure [83,84]. These are classical modifications shared by class A GPCRs, as reviewed in [85]. Moreover, the Gs proteins in complex with V2R adopted a conventional architecture that is shared with other class A GPCRs, as reviewed in [70]. Upon activation by the natural hormone, the allosteric rearrangement of the TM5-TM6-TM7 intracellular side allowed the accommodation of the G protein α_5_ helix insertion into the receptor bundle and made molecular contacts with TM1, TM2, TM3, TM5, and TM6. ICL3 was not observed in any of the structures due to its size and intrinsic dynamic. The ICL2 was only visible in the EM map using the nanobit approach that artificially stabilized the interface [81] and where the loop interacted with the α5 helix, the αN helix, and the β1 strand of the G_S_ protein RAS-like domain. Interestingly, the wild-type Gs protein structures differed by a tighter and more dynamic interface with potential molecular contacts between V2R ICL1 and the blade 7 loops of the Gβ subunit [79]. Since the miniGs protein engineering may introduce bias into the V2R-G–protein interface, it is interesting to combine the data from these three studies, taking into account their respective limitations to provide a more complete picture of AVP-V2R-Gs’ protein dynamics and structure. The recent development of new tools to probe protein flexibility within a cryo-EM dataset for both the interdomain motion [86] and continuous variability [87,88,89] allowed the assessment of AVP-V2R-Gs’ protein dynamics. This multi-body refinement performed a principal component analysis (PCA) on the respective positions and orientations of user-defined domains of the analyzed particles and described the variability by a set of eigenvectors and eigenvalues. Indeed, a multi-body refinement was performed on the complex with the wild-type Gs protein structure to assess the relative motion between the V2R and the Gs protein heterotrimer. A strong dynamic was probed for the complex, which is consistent with what was expected with the use of wild-type G proteins.

Subsequently, the structure of the V2R complex with βarrestin1 (βarr1) was also characterized by cryo-EM (Figure 6b) [30]. Despite their central role in GPCR trafficking and signaling regulation [90], the βarrestin-GPCRs interaction is still poorly understood at the molecular level, and only a handful of structures have been characterized so far [91]. This is mainly due to the properties of binding βarrestin to GPCR, which make these complexes particularly dynamic. Indeed, this mechanism is driven by two main interactions: first, the phosphorylated pattern PxPP [92,93,94], either located in the ICL3 or the C-terminal area and presented in most GPCRs, binds to the N-Lobe of βarrestin and disrupts the “three-element” interaction, driving βarrestin activation and leading to the “hanging conformation” [95]. Thus, active βarrestin can interact with both the receptor core and the lipid bilayer, inducing further conformational changes [96,97,98] and resulting in “core conformation”. Currently, all the reported structures of βarr1 in complex with either the M2R, the β1-adrenergic receptor (β_1_AR), the neurotensin NTSR1 (reviewed in [99]), the serotonin receptor 2B (5HTR_2B_) [28], and the V2R have been represented in their core conformation. The V2R receptor and the βarr1 both adopt active conformation with the AVP and adopt the same orientation as those of the V2R-Gs protein structures discussed previously, while the average resolution of 4.7 Å prevents an atomic-scale interpretation and the observation of subtle changes in the receptor compared to the Gs protein complex, where observations on the interface and of the βarr1 side were possible. Interestingly, the finger loop of βarr1 inserts into the receptor core in the same overall location as the Gs α_5_ helix in the AVP-V2R-Gs structures. Additionally, all V2R ICLs interact with the βarr1:an interface that may explain the strong interaction between V2R and βarr1. Moreover, the V2R–βarr1 interface is significantly different from those of the M2R–βarr1, β_1_AR–βarr1, and NTSR1–βarr1 complexes but similar to the recently reported 5HTR_2B_–βarr1 complex. Indeed, with respect to the position of the receptor, V2R-βarr1 has been shown to adopt an intermediate orientation compared to the two extremes of the β1AR-βarr1 and M2R-βarr1 structures on the one hand and the NTSR1-βarr1 structures on the other. In these structures, βarr1 also displayed variable tilts towards the membrane depending on the reconstituted system (detergent vs. nanodiscs) and the presence or absence of the lipid phosphatidylinositol (PI(4,5)P_2_). Interestingly, this phenomenon has been proposed to be physiologically relevant in the case of strong membrane curvatures, such as during clathrin-mediated internalization [99]. This GPCR–βarrestin interface plasticity provided a basis with which to explain how only two βarrestins (βarr1 and βarr2) can interact with hundreds of GPCRs and finely tune their signaling and trafficking.

## 5. Solution NMR and Molecular Dynamics—Key Tools to Decipher the Dynamic of GPCRs

X-ray crystallography and cryo-EM structural studies have provided important insights into the process of receptor activation. However, these represent snapshots of a highly dynamic process. In this respect, NMR spectroscopy in solution is a useful method for analyzing function-related conformational equilibria in GPCRs, as reviewed in [8,100,101,102]. This chapter aims to summarize the complementary information obtained by solution NMR with a particular focus on µOR and V2R as they relate to allosteric coupling, variable efficacy, and the biased signaling of GPCR ligands, which are of particular interest for their potential as drugs.

### 5.1. Solution NMR on µOR

As mentioned above, crystal structures of µOR have been solved in both inactive (bound to antagonists [58]) and active states (the ternary complex form of µOR, full agonist, and a G protein-mimetic nanobody [64]). At the time, these crystal structures were complemented by NMR studies using the advantage of the versatility of NMR spectroscopy, as described for β2AR [34]. To study the receptor by solution NMR, the mouse µOR, the same construct used in the crystal structure of the ternary complex [64], was expressed in *Sf9* insect cells, reconstituted in the LMNG micelle and finally in the post-translational reductive ^13^C-methylation to methylate the ε-NH2 groups of lysine side chains (ε-N[^13^CH_3_]2-lysines); these were used to analyze the activation signal propagation that occurred in different receptor domains [66].

The [^13^C, ^1^H]-heteronuclear multiple quantum correlations (HMQC) spectra of the ^13^C labeled µOR showed peaks with different intensities in the unbound state. The NMR peak assignments were obtained by single-point mutagenesis of an individual lysine residue, and NMR sensors were divided into two groups: extracellular and intracellular. Together, this observation qualitatively indicated that different domains that are intracellular present different dynamic properties, that is, K98^ICL1^, K100^ICL1^, and K344^8.51^ for (ICL1/H8) and K269^6.24^/K271^6.26^ for TM6. Comparisons of the spectra of µOR with three different agonists (BU72, DAMGO, and [Dmt^1^]DALDA), when bound, indicated that the binding of the G protein mimetic nanobody was required for the agonist to fully stabilize an active conformation in both extracellular and intracellular domains, suggesting the existence of a two-way allosteric coupling between the extracellular µOR ligand-binding domain and the G protein–coupling interface. The quantitative analysis of intracellular NMR sensors suggested that allosteric coupling from the agonist-binding pocket to ICL1/H8 was stronger than their coupling to TM5 and TM6. These results offered insights into a possible allosteric pathway of activation from the agonist-binding pocket to the ICL1 and H8 domains to initiate the ternary complex formation and associated changes in TM5 and TM6. It was easy from these results to speculate that a G protein could first engage ICL1 and/or the H8 region before fully engaging with TM5 and TM6, even if the NMR experiments did not provide the temporal sequence information of these conformational changes.

Methods such as X-ray crystallography and cryo-EM that capture the lowest energy states within an ensemble of conformations do not provide much information about the conformational dynamics of GPCRs, which is crucial to fully understanding the involvement of drug binding, receptor conformational changes, and subsequent interactions with cellular signaling partners. Such complexity is critical for the GPCR function as it likely drives many aspects of ligand pharmacology, including the ability of some agonists to preferentially activate G protein signaling over β-arrestin recruitment [103]. Indeed, all the recent cryo-EM structures of µOR, obtained with either unbiased or biased agonists [72,73,74], revealed some of the binding poses of agonists on G protein stabilized structures (Figure 4). It should be noted that NMR studies demonstrated the existence of two-way allosteric coupling; the G protein certainly overwhelmed the effects of the ligands on the intracellular side of the receptor and also in the ligand-binding pocket on the extracellular side.

Interestingly, conformational states that remain unobserved by high-resolution structural methods were first demonstrated in a NMR study of human µOR using a ^13^C methyl-specific labeling scheme in a highly deuterated background in the insect cell expression system that was previously established [36]. The NMR resonances of the methionine residues in µOR/∆6M, where six methionines were mutated to simplify the spectra, revealed that µOR existed in a conformational equilibrium [104]. Indeed, M^5.49^ was close to the PIF motif and underwent a conformational change upon the activation of µOR. Indeed, two resonances of M^5.49^ were observed. The relative intensities between the two resonances correlated well with the efficacy of the agonists. M^3.46^, M^5.61^, and M^6.36^ of µOR were sensors of conformational changes on the intracellular side of TM3, TM5 and TM6, respectively. The NMR signals of these methionine residues demonstrated that the G protein–biased agonist (oliceridine) induced a different conformation of µOR than the unbiased agonist DAMGO. Altogether, these results suggest that the conformational equilibrium observed for M^5.49^ propagated to the intracellular cavity of µOR and existed in an equilibrium between close and open conformations in a ligand-dependent manner.

Most importantly, these NMR studies have shown that µOR exists in a conformational equilibrium in solution, with agonists and antagonists stabilizing unique active or inactive states, respectively. Interestingly, a follow-up study from the same group [105] used the same methyl-specific labeling scheme to analyze µOR conformational equilibrium in the presence or absence of BMS-986122 with or without an orthosteric agonist. Indeed, BMS-986122 is a positive allosteric modulator (PAM), which means that it is a ligand that binds to non-orthosteric sites to enhance the signaling activities of GPCR [106]. Following the signals from M^5.61^ and M^6.36^, which probe, respectively, the intracellular side of TM5 and TM6 revealed that µOR exists in a conformational equilibrium. Using various orthosteric ligands with or without the PAM BMS-986122 shift, this equilibrium could be correlated to the signaling activity of µOR. At least three different conformations were probed with different activities. The population of these conformations defined the apparent signaling activity on the intracellular side of the µOR. The addition of PAM enhanced µOR activity by increasing the population of the fully activated conformation in the conformational equilibrium to a level that could not be reached by orthosteric ligands alone. In agreement with the previous study [66], this NMR study showed that the activity of µOR in the full agonist DAMGO-bound state did not reach its potential maximum because the inactivated and partially activated conformations were significantly populated in this state. Moreover, the perturbation of M^3.36^, M^3.46^, and M^5.49^ NMR signals upon the addition of BMS-986122 indicated that the local environments of these residues were in the vicinity of the PAM binding site. To identify the BMS-986122 binding site, methionine residues were introduced by point mutations near these residues in TM3, TM4, and TM5 at the positions of W^3.41^, L^4.92^, L^5.38^, and I^5.44^. Only the NMR signal of F158M^3.41^ and L196M^4.52^ variants exhibited chemical shift differences and considering that BMS-986122 has been shown to exhibit subtype selectivity for µOR over the other opioid receptor subtypes δ and κ where T^3.45^ is a non-conserved residue among the other opioid receptors, both have a methionine residue in the corresponding positions. Altogether, these results suggest that BMS-986122 recognized a cleft in the surface of the transmembrane region, and T^3.45^ resided at the bottom of this cleft.

As the NMR studies mentioned above thus reveal, µOR and class A GPCRs are not static on/off switches but complex molecular machines that operate through strictly regulated motions. Ligands may preferentially activate or inhibit distinct signaling pathways by changing the conformations of the GPCR. This is known as functional selectivity providing fine regulations of GPCR signaling and relies on the dynamic equilibrium of GPCR conformations. To further understand the functional selectivity of µOR, it is important to identify specific µOR conformations that could lead to biased signaling. A dual-isotope methyl-labeling NMR scheme was implemented, on methionines and lysines, to simultaneously monitor the extracellular binding site, the intracellular coupling region, and the connector region. The combination of these NMR data with the advanced REST2 enhanced-sampling MD simulations was used to capture conformational dynamic patterns during the pre-activation stage, which differentiated the partial (buprenorphine) and the biased agonists (PZM21, oliceridine) from the fully unbiased ones (BU72, DAMGO) [107]. In detail, the binding of unbiased agonists caused the toggle switch W^6.48^ in TM6 to approach TM2 in the connector region. Instead, biased agonists bound deeper in the pocket and separated TM6 from TM2 in the connector region, allowing TM7 to approach TM3. This perturbation triggered conformational changes in TM7, ICL1, and H8. This occurred allosterically via the conserved motifs N^7.49^P^7.50^xxY^7.53^ in the TM7, D^2.50^ at the Na^+^-binding site, and G^1.49^N^1.50^ in TM1. Thus, specific µOR conformations leading to biased signaling have been identified. Specifically, the lower half of TM7 moved toward TM3, closing the cleft between H8 and the ICL1, and these distinct TM7-ICL1-H8 conformations could inhibit the binding of β-arrestins but not the G proteins. This biased mechanism involved conserved motifs in class A GPCRs, which may also apply to other GPCRs.

### 5.2. Solution NMR on V2R Receptor, Cut Complementary Information

The characterization of large membrane proteins, such as GPCRs, remains challenging by NMR while allowing fundamental insights into GPCR from their dynamics, allostery, and their conformational rearrangements. Indeed, despite the recent V2R structures, some structural domains were not resolved in the cryo-EM structure [30,79,80,81], such as ICL3 and the C-terminal, mostly due to their intrinsic dynamics. Importantly, ICL3 is required for coupling with its G protein effector [108], and the arrestin interaction requires a phosphorylated C-terminal domain with the so-called phospho-barcode model [109,110]. However, NMR studies on these truncated receptor regions, the ICL3 [32], and the C-terminal domain [33] have contributed to our understanding.

To obtain further insight into the structure–dynamics–activity relationship of ICL3, either isolated or in complex with target proteins, it was necessary to produce large quantities of the isotopically labeled (^15^N or ^15^N–^13^C) recombinant peptide [111]. The overexpression and purification of such truncated regions are not straightforward since affinity tags, often tolerated for larger proteins, have to be removed to prevent artifacts in the structural study. To reproduce the hairpin fold that is presumably present in the full-length receptor, a disulfide bond between the N- and C-terminal extremities was introduced. The ICL3 domain, corresponding to V2R fragment residues C224^5.57^-T273^6.37^ with an extra cysteine in C-terminal, was overexpressed as a fusion protein with thioredoxin A at the N-terminus and a hexahistidine tag. After purification and in the presence of DPC micelles mimicking the plasma membrane, ICL3 adopted a left-twisted α-helical hairpin conformation, where the N- and C-terminal parts formed well-defined packing helices that extended about 17 amino acids each, TM5 and TM6. The original feature of these studies [32,111] was, at the time, the length (50 amino acids) that showsedICL3 to adopt a well-defined structure with some conformational flexibility that was not dependent on the length of ICL3.

As is the case for many C-terminal domains of GPCRs, the C-terminal region of V2R was shown to be disordered. To characterize this domain experimentally, the C-terminal region of V2R (343-371) was expressed and purified [33]. NMR was used to probe the conformational and dynamic preferences at the residue level, using a set of complementary experiments, such as secondary chemical shifts (SCS), scalar (J) and residual dipolar couplings (RDCs), paramagnetic relaxation enhancement (PRE), and relaxation [112,113,114]. The C-terminus of V2R displayed a central helix from residues E356 to S364. Moreover, the propensity of this secondary structure was low, which illustrates the high flexibility of the C-terminal region and its ability to adopt distinct conformations in the solution. However, it is interesting to note that their residual secondary structures encompass residues that are known to be phosphorylated by GRKs [115] and have been resolved in a cryo-EM structure [30] that adopts a β-strand conformation and interacts with the β-arrestin1. This result suggests that a conformational change must occur upon binding to β-arrestin1 and/or phosphorylation. This structural transition might be at the basis of the GPCR:arrestin interaction.

## 6. Concluding Thoughts

The identification of new GPCR drugs requires detailed knowledge of GPCR biology, particularly structural biology, due to the complex structure–function relationships involved in GPCR signaling. More GPCR structures have been determined in the last four years (2020–2023) than in the previous twenty years (2000–2019). Most of these are GPCRs in complex with heterotrimeric G proteins, which have become relatively routine today thanks to their large size for single-particle cryo-EM structure determination and the use of tools that rigidify either GPCRs and/or nucleotide-free G proteins (miniG, nanobody, scFv, Fab). However, complexes with other GPCR effectors, such as β-arrestins and GRKs, remain challenging due to their high intrinsic dynamics, conformational heterogeneity, and the multiple modes by which they can interact with activated receptors. Although confounding in terms of their understanding, the observed conformational heterogeneity is functionally relevant because it likely allows them to adapt to many different receptor and membrane contexts. Compared to X-ray crystallography, the incorporation of cryo-EM results is an exciting prospect. Indeed, cryo-EM has a much greater potential to provide structures of multiple conformations simultaneously in a given sample preparation. Depending on the question, if the goal is to observe the exact positioning of a specific ligand, several methods are now available to counteract the dynamics of a given receptor. On the other hand, if the goal is to understand the dynamics of the protein, it is better to characterize a protein that is as close to its physiological state as possible. Some cryo-EM software now offers the ability to determine the dynamics of the protein at an intermediate resolution. In order to successfully understand a given GPCR, it is essential to juggle these different possibilities to deeply decipher the mechanism occurring at the GPCR level. Indeed, dynamic studies of GPCRs are important to provide new insights into GPCR biology that can support structure-based drug design. It is exciting to look forward to the use of integrative GPCR structural biology, such as combining the information from cryo-EM structures of multiple conformations and the potential of solution NMR to analyze function-related conformational equilibria in GPCRs as they relate to allosteric coupling, variable efficacy, and the biased signaling of GPCR ligands, which are of particular interest for their potential as drugs. Every advance in structural biology could aid in the discovery of new drugs, as well as the improvement of existing drugs and their application. A critical goal for the future is to predictively design not only the desired signal selectivity but also the molecular properties needed to advance molecules into the clinic.

## Figures and Tables

**Figure 1 membranes-13-00606-f001:**
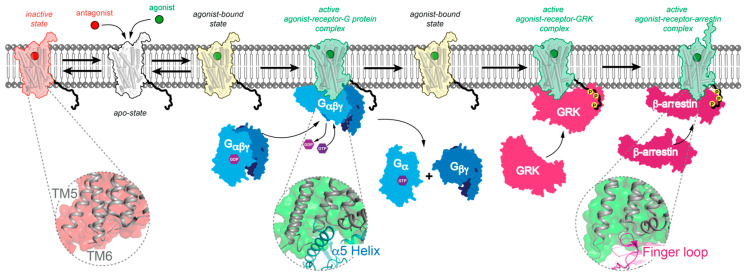
Schematic representation of GPCR activation. Prior to ligand binding, the receptor is in an apo-state (receptor is in white). Upon the binding of an antagonist (red circle) to a receptor, there is no activation (receptor is in red). Upon the binding of an agonist (green circle), the receptor is in the pre-activation state (receptor is in yellow), and the G protein heterotrimer binds to the active receptor (active receptor is in green). The exchange of GDP for GTP in the G protein α subunit leads to dissociation and interaction with downstream effectors such as the Gα subunit with the adenylyl cyclase (AC) and Gβγ subunits that activate ion channels. Pre-activated receptors can also signal through arrestins. The phosphorylation of the receptor C-terminal tail (yellow circle) by G protein-coupled receptor kinase (GRK) binding (active receptor is in green) promotes arrestin recruitment (active receptor is in green), which can internalize the phosphorylated receptor. Gray dashed circles highlight the common binding pocket of the receptor where the α5-helix (blue cartoon) of the Gα subunit and the finger loop (purple cartoon) of arrestin bind upon the activation of the receptor (grey cartoon). The G proteins, GRK and arrestin are shown in blue, pink and purple, respectively. The apo, inactive, agonist-bound and active states of the receptor are shown in white, red, yellow, and green, respectively.

**Figure 2 membranes-13-00606-f002:**
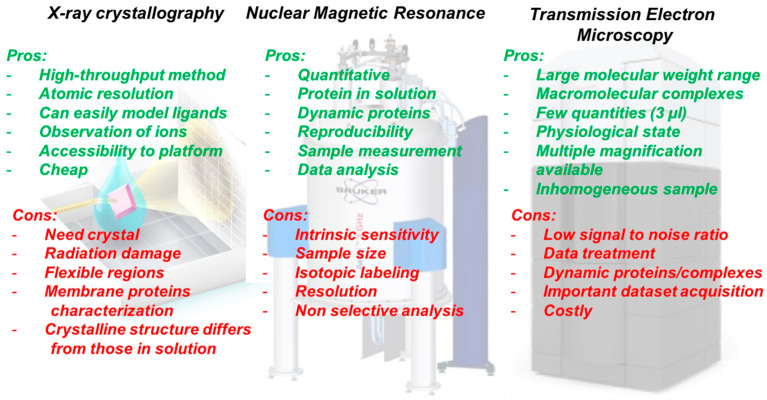
Comparison of the pros and cons of the main structural biology methods.

**Figure 3 membranes-13-00606-f003:**
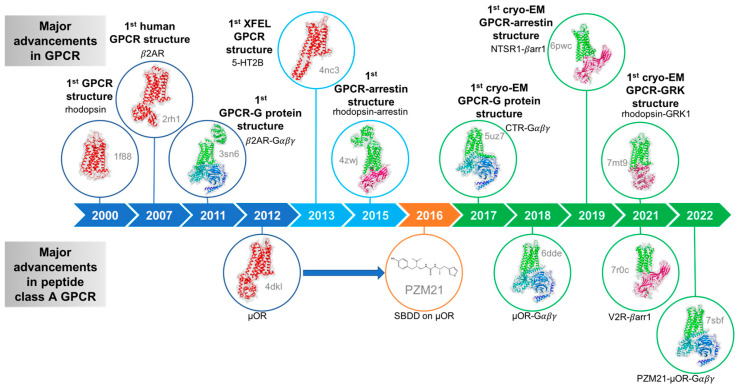
Timeline of major advancements in the GPCR field. The achievements of X-ray crystallography, structure-based drug design and cryo-EM are shown in blue (light blue for XFEL), orange, and green, respectively.

**Figure 4 membranes-13-00606-f004:**
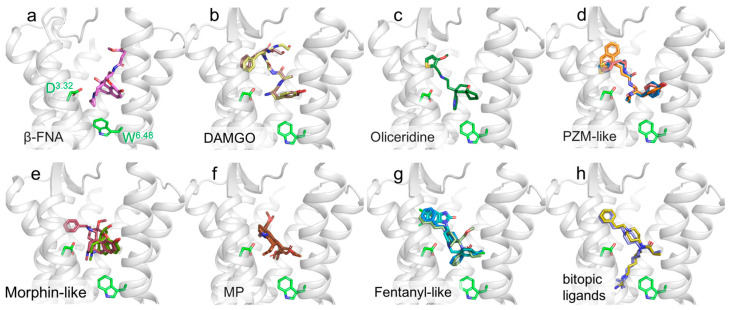
Close-up view of the µOR binding site. µOR structures are shown as gray cartoon representations, D^3.32^ and W^6.48^ are displayed as green stick representations. (**a**) β-FNA is in violet (4dkl); (**b**) DAMGO is in light yellow (6dde) and light brown (8efq); (**c**) oliceridine is in dark green (8efb); (**d**) PZM21 is in blue (7sbf) and pink (8efo) and FH210 is in orange (7scg); (**e**) BU72 is in raspberry (5c1m) with morphine in green (8ef6); (**f**) MP is in brown (7t2g: 2 poses). (**g**) fentanyl in blue (8ef5), SR17018 in cyan (8efl) and LFT in light green (7t2h); (**h**) C5 guano is in blue (7u2l), and C6 guano is in gold (7u2k).

**Figure 5 membranes-13-00606-f005:**
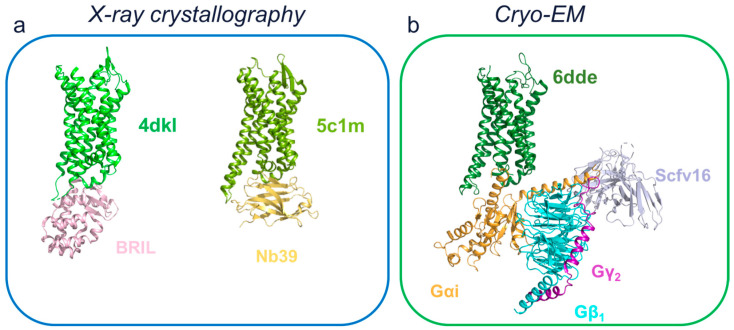
Structures of µOR in different states. (**a**) X-ray structures of the inactive-(left) and active-state (right) µOR bound to antagonist β-FNA and agonist BU72, respectively. (**b**) Cryo-EM structure of µOR in complex with the DAMGO and Gi protein. All proteins are shown as cartoon representations and ligands are not shown. µOR are depicted in a green color range. Gαi subunit is shown in orange, Gβ subunit is shown in cyan, Gγ subunit is shown in purple. BRIL, Nb39 and ScFv16 are illustrated in pink, yellow and light blue, respectively. The PDB file accession numbers are indicated for each complex.

**Figure 6 membranes-13-00606-f006:**
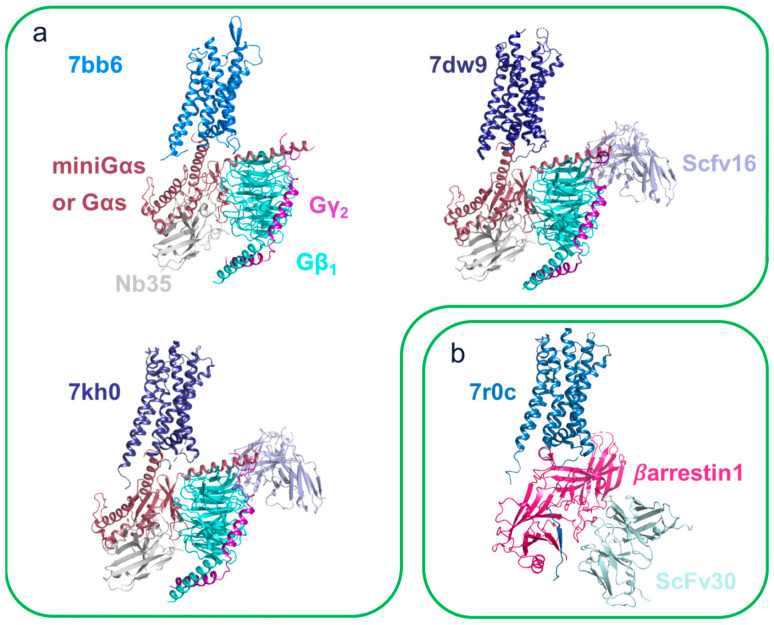
Structures of V2R complexes. (**a**) Cryo-EM structures of V2R with Gs protein. (**b**) Cryo-EM structures of V2R with β-arrestin1. All proteins are shown as cartoon representations and AVP are not shown. V2R are depicted in the blue color range. The Gαs subunit is shown in brown (mini Gαs in 7kh0 and 7dw9, complete Gαs in 7bb6), the Gβ subunit is shown in cyan, and the Gγ subunit is shown in purple. Nb35 and ScFv16 are illustrated in gray and light blue, respectively. β-arrestin1 and ScFv30 are shown in pink and pale cyan, respectively. The PDB file accession numbers are indicated for each complex.

## Data Availability

Not applicable.

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
