# Peer review of "Bringing GPCR Structural Biology to Medical Applications: Insights from Both V2 Vasopressin and Mu-Opioid Receptors"

_membranes, 2023, doi:10.3390/membranes13060606_

Round 1

Reviewer 1 Report

The authors present a review about structural studies of two GPCRs, vasopressin receptor (V2R) and the mu-opioid receptor (uOR). Structures of GPCR in complex with various proteins and ligands resolved by X-ray crystallography and cryo-EM were compared and discussed in detail. Those structures are static snapshots of dynamic GPCR structure, so in-depth review of NMR and molecular dynamics simulation studies of GPCR was also included. The presented information shed lights on the mechanism of GPCR activation and provide strong supports for the development of drugs targeting V2R and uOR via SBDD. Overall, this manuscript is well organized and would be helpful for the drug design targeting GPCR.  However, considering the following issues, the manuscript is not suitable for publication before revisions. 

Major:

1.     Although “show the impact of this integrative approach on the determination of new potential clinical effective compounds” is one important objective of this review, it is not clear how previous ligands targeting GPCR were developed, and how the X-ray/cryo-EM/NMR structures would benefit drug design. Previous SBDD studies on V2R and uOR should also be reviewed.

 Minor:

1.     Page 5, line 205, “endothelium [41]or”  is a typo of “endothelium [41] or”.

Reviewer 2 Report

Fouillen and Bous and colleagues provide an informative review on two GPCRs. However, this review has lots of room to improve. As an expert on structural biology, I do not find that description of method development section was written with clarity as it could be.

Here is my suggested outline:

(1) what is this review about, why GPCRs, what methods, related human diseases,

(2) update on experimental methods development,

(3) two mu-OR examples (why these two),

(4) direct visualization of GPCRs by X-ray crystallography and cryo-EM,

(5) inferences from NMR spectroscopy,

(6) visualization by computational methods

(7) Concluding thoughts or summary.

Since (4) and (5) describes a general history in addition to mu-OR, it is unclear where and why to place these two specific examples (3) in the context of this review. The review should be written in a logistics that is easily to follow for scientists outside the field.

Lines 168-172:

This review aims to present an updated knowledge of the structural tools that have enabled the structural and dynamic characterization of GPCRs based on the case of two widely drug-targeted class A GPCR receptors, the mu-opioid receptor (μOR) and the arginine-vasopressin (AVP) V2 receptor (V2R).

This statement is clearly out of the place. It should be at the beginning of this manuscript somewhere so that reader will know in advance what the authors are going to tell the reader about this review.

Likewise, line 174-175 is also out of the place and should be at the begin of this review.

This review will focus on the two class A GPCRs responding to small peptides.

More precisely, they are two small-peptide responding class A GPCRs.

Before Section 2, a schematic drawing would be helpful on the issue what is considered to be active and what is inactive, which is highly relative terms, and how the GPCR pathway works in general, what is agonist and antagonist. For a reader who are not working on this area, these terms can be very confused. For example, the receptor-arrestin complex is active for resetting the system whereas it is active for receptor-coupled phosphorylation.

A table or diagram of historical events might be helpful for describing major advancements in the GPCR field.

When citing Ballesteros and Weinstein nomenclature, one or two sentence description of their nomenclature would be helpful so that scientists outside this field would not need to look at and read the cited papers. The nomenclature starts with the first transmembrane number, 1 to 7, followed by residue numbers flanking the most conserved residue number being set 50.

I found that the methods described in Bous et al. (2021) Sci Adv paper was very interesting to the structural biology community in which authors identified E1, E2, E3, and E4 four conformations shown in their Figure 1 to demonstrate flexibility in the AVP-V2G-Gs-Nb35 complex. There was no citation or description in that paper as to how these eigenvalues were determined. Perhaps, authors could add a few words on the missing information in the earlier publication.

Please label Galpha, Gbeta, Ggamma, and arrestin inside Figure 2. I have some difficulties to tell differences between purple and pink colors when figure sizes are so small.

slightly above average

Round 2

Reviewer 1 Report

The manuscript has been sufficiently improved. I think it is appropriate for publishing in Membranes.

Reviewer 2 Report

This revision has significantly improved for non-GPCR specialist readers. However, all figures in PDF version of manuscript have very low resolution. Hopefully, the high-resolution figures will be published in the final version.

It was about average